# Truth-Driven Negative Sampling in Self-supervised Graph Representation Learning

## Abstract

Negative sampling is an important yet challenging component in self-supervised graph representation learning, particularly for recommendation systems where user-item interactions are modeled as bipartite graphs. Existing methods often rely on heuristics or human-specified principles to design negative sampling distributions. This potentially overlooks the usage of an underlying "true" negative distribution, which we might be able to access as an oracle despite not knowing its exact form. In this work, we shift the focus from manually designing negative sampling distributions to a method that approximates and leverages the underlying true distribution. We expand this idea in the analysis of two scenarios: (1) when the observed graph is an unbiased sample from the true distribution, and (2) when the observed graph is biased with partially observable positive edges. The analysis result is the derivation of a sampling strategy as the numerical approximation of a well-established learning objective. Our theoretical findings are also empirically validated, and our new sampling methods achieve state-of-the-art performance on real-world datasets.

## 1 Introduction

Self-supervised graph representation learning has wide applications in modern recommendation systems. To obtain high-quality embeddings for users and items, the typical method is to model past user-item interactions as a bipartite graph. Node embeddings are often learned in such a way that the distance between a user's embedding and an interacted item's embedding indicates the likelihood of an edge between the two objects.

Negative sampling is an important step in this learning paradigm: for a given anchor node $u$, a node $v^-$ needs to be sampled from a certain distribution, so that node pair $(u, v^-)$ can be treated as a "negative edge" in training, together with the observed positive edge $(u, v^+)$, for training the model via binary loss. Negative sampling is important because it generates all negative samples in the training dataset by some human-specified rules, which can have much influence on learned representations. This also applies to more general graphs arising outside recommendations, which is also considered in our discussion hereafter.

Negative sampling is not only important but also challenging. Existing works have proposed many heuristics, including random negative sampling (RNS) (Rendle et al., 2012), popularity-based negative sampling (PNS) (Mikolov et al., 2013), hard negative sampling (HNS) (Ying et al., 2018), GAN-based negative sampling (GAN-based NS) (Chae et al., 2018), and in-batch negative sampling (in-batch NS) (Wu et al., 2021), *etc.*. More recent theoretical studies (Yan et al., 2024; Yang et al., 2020b) have proposed properties of node embeddings that a good negative distribution should ideally encourage to learn, and then design the negative distributions accordingly.

Should we consider negative sampling to be primarily a matter of human design, or could there be a more grounded and established objective which we can rely on so that negative sampling can be more fundamentally viewed (and optimized) as a numerical process to approach that objective?

In particular, we want to start by formulating and highlighting the concept of the true distribution of negative samples $p^-$, whose objective existence is independent of human choices. As a motivating example, consider that we are in an ideal world where we have enough resource and capability to survey the true opinion of each user $u$'s potent to like each item $v$ in the recommendation

pool. This true opinion value can be technically formulated as $P(y = 1 | e = (u, v))$, where $y$ is the true class label of the user's like. By applying Bayes rule, we can quickly get distribution $p^+ = P(e | y = 1) \propto P(y = 1 | e) P(e)$ and distribution $p^- = P(e | y = 1) \propto (1 - P(y = 1 | e)) P(e)$, where $P(e)$ is the edge prior. In the above formulation, $p^-$ is the true negative distribution that to emphasize: every graph from a well-defined domain in practice should have such a $p^-$ that objectively exists. We will provide more rigorous definitions in Section 3.

Of course, the exact form of $p^-$ is mostly assumed unknown, otherwise the classification problem is already algebraically solvable by reversely deriving $P(y = 1 | e)$ from $p^-$, in which case there is no need to do sampling and training. However, $p^-$ being unknown does not necessarily preclude us from being able to access (samples from) $p^-$ as an oracle (though again in many cases there is distribution shift in observation, which we will have a dedicated section to address). The reason that we may still need to generate negative samples from a different distribution than $p^-$ is primarily to reduce variance of learning, *i.e.* to speed up convergence with fewer samples. Some existing analysis (Mukherjee et al., 2013) of classical boosting algorithms such as AdaBoost helps further illustrate this point.

Formulating $p^-$ guides the derivation of a good *proposal* negative distribution $q$. A good $q$ should be grounded and focused on leveraging the noted true (albeit unknown) negative distribution $p^-$ — including understanding how $p^-$ factors into the expression of a well-established optimization goal, as well as how it can be approximated or recovered if presumed distorted. In fact, an interesting observation is that most existing negative sampling methods made this assumption implicitly without fully exploiting its usage. See Appendix A for more discussion.

**Our work.** In this work, we analyze how to build towards an good sampling strategy, by shifting the focus from proposing distributions into crafting sampling procedures based on the argument and the leverage that a true $p^-$ exists, which further expands into two cases. In the first case, we assume that the observed graph is an unbiased sample from the true distribution. $q^-$ can then be derived by drawing its connection to a well-established objective that involves $p^-$. It turns out that the resulted sampling strategy can be interpreted as a form of adaptive hard negative sampling.

In the second case, we assume that the observed graph is a biased sample from the true distribution, and that the positive edges may be only *partially* observable. This case is underexplored in graph learning's literature but has wide applications. For example, in recommendations a positive edge represents a user's manifested interest in an item logged by the platform; however, the absence of an edge can either mean the user truly dislikes the item, or that the user just has not been offered the chance to interact the item but would have liked it if so. In fact, due to the sparsity nature of many complex systems only a small fraction of positive edges may be observable.

To address the second case, the idea is to first derive an unbiased empirical risk estimator depending on *true* distributions, and then convert it into a form that depends only on *observed* distributions, up to some calibration. The calibration, which was deemed a bottleneck in more general settings, can be facilitated by graph topology in a learnable manner. We further validated this new sampling method on real-world datasets and achieved state-of-the-art results.

**Contributions.** In summary, we make the following contributions.

- We propose to fundamentally shift the mode of thinking from proposing negative distributions to approximating and utilizing the true distributions in sampling methods for graph representation learning in both recommendations and beyond.

- We theoretically derive and analyze the forms that the optimal sampling strategy should follow, under the assumption that the positive edges are sampled unbiasedly from the true distribution, or that the positive edges are only partially observable.

- We further validate our theorems and new sampling method on real-world data and observe their significant improvement over previous methods.

## 2   RELATED WORK

**Graph Representation Learning.** Graph representation learning (GRL) studies how to represent relational information in low-dimensional vector spaces. Self-supervised GRL aims to train a graph/node encoder using the graph data itself without extra labels, which further splits into two categories: generative learning, whose supervision signals purely come from the data, and contrastive

learning, whose supervision signals come from both the original data and some stochastically augmented data. Graph contrastive learning has many applications including recommendations (Shuai et al., 2022), drug discovery (Wang et al., 2021), anomaly detection (Wang et al., 2024), *etc.*.

For scalability and efficiency, fancy data augmentations are less often seen in graph contrastive learning for industry-level recommendations (Yu et al., 2022). In recommendations, positive samples are usually directly sampled from the edge set or random walks (Ying et al., 2018), and negative samples by fixing an anchor node then drawing the second node from a certain negative distribution Yang et al. (2020a). Popular loss terms include cross entropy loss (Yang et al., 2020b), Bayesian personalized ranking loss (Rendle et al., 2012), and InfoNCE loss (Oord et al., 2018).

**Negative Sampling.** Negative sampling was originally proposed as a component in noise contrastive estimation (Gutmann & Hyvärinen, 2010; 2012), and was soon applied to language modeling (Mikolov et al., 2013; Mnih & Teh, 2012) for approximating softmax. It emerged in graph's context when node embedding methods (Perozzi et al., 2014; Grover & Leskovec, 2016) took inspirations from language modeling. Since then, it has been widely applied in self-supervised graph learning, and in recommendations (Yang et al., 2020a), and soon became a standalone research topic due to its importance.

Existing works on negative sampling have proposed many heuristics, including random negative sampling (RNS) (Rendle et al., 2012; Bordes et al., 2013), popularity-based negative sampling (PNS) (Mikolov et al., 2013; Perozzi et al., 2014), hard negative sampling (HNS) (Ying et al., 2018; Zhang et al., 2019; Huang et al., 2021), GAN-based negative sampling (GAN-based NS) (Chae et al., 2018; Wang et al., 2018), and in-batch negative sampling (in-batch NS) (Wu et al., 2021; Chen et al., 2020; Zhao et al., 2021), *etc.*. More recent theoretical studies (Yan et al., 2024; Yang et al., 2020b; Robinson et al., 2021) proposed properties that a good negative distribution should satisfy, such as monotonicity and accuracy, and then make designs accordingly. Shi et al. (2023) further analyzes the relationship between hard negative sampling and BPR loss.

# 3 PROBLEM FORMULATION AND PROBABILISTIC SETUP

## 3.1 PROBLEM FORMULATION

We consider the standard task of self-supervised learning on a graph $G = (V, E)$ where $V$ is the node set, $E$ is the edge set. Denote by $x_u \in \mathbb{R}^m$ the raw features of node $u \in V$. The learning goal is to obtain a graph encoder $f$ defined as a mapping $\mathbb{R}^m \to \mathbb{R}^n$, so that $f(x_u)$ can best represent node $u$ and be used in downstream tasks. $f$ is usually parameterized by a neural network with parameters $\theta$, learned through minimizing the following objective:

$$J = \sum_{\substack{(u,v)\in E, \\ v_1^-,\ldots,v_k^-\sim q^-}} \left[ L^+(u,v) + \frac{\tau}{k} \sum_{i=1}^k L^-(u,v_i^-) \right] \tag{1}$$

$$\text{where} \quad L^+(u,v) = g^+(f(x_u), f(x_v)), \quad L^-(u,v^-) = g^-(f(x_u), f(x_{v^-})) \tag{2}$$

which intuitively traverses every edge $(u,v) \in E$, picking $k$ negative nodes from the proposal negative distribution $q^-$ for each edge, and collects the corresponding positive and negative losses. $g^+$ and $g^-$ are simple functions that measure dissimilarity and similarities between the $f$-encoded node embeddings $f(x_u)$ and $f(x_v)$, respectively. $\tau$ is a weighing constant for analysis purpose, which we can assume to be equal to 1 by default.

The negative sampling problem asks:

($*$) *What is the best form of $q^-$ to use in the above objective $J$?*

This problem definition is consistent with the one in Yang et al. (2020b); Yan et al. (2024) though we will take a distinct path towards its solution. Notice that

**Proposition 1.** *J is the empirical estimation of the following expected risk term*

$$R_1(f) = \mathbb{E}_{u\sim p^+}[\mathbb{E}_{v\sim p^+(\cdot|u)}[L^+(u,v)] + \mathbb{E}_{v^-\sim q^-}[L^-(u,v^-)]] \tag{3}$$

where $p^+ : V \times V \to [0, 1]$ is the underlying positive distribution from which all edges in $E$ are drawn. As a notation convention in this paper, we use $(u, v) \sim p^+$ to mean that $u, v$ are drawn from the joint distribution of $p^+(u, v)$, while $u \sim p^+$ means that $u$ is drawn from the marginal distribution $p^+(u)$.

## 3.2 PROBABILISTIC SETUP

To provide a more rigorous definition for $p^+$ and various other distributions, we consider probabilities in the sample space $\{(e, y, s)\}$: (1) Random variable $e = (u, v) \in V^2$ is a node pair in the graph. (2) Random variable $y \in \{0, 1\}$ is the class indicator variable for ground truth. $y = 1$ means that the associated node pair $e$ is a true positive edge, and $y = 0$ otherwise. (3) Random variable $s \in \{0, 1\}$ is the observation indicator variable. $s = 1$ means the edge is observed, and $s = 0$ otherwise.

Based on the above definitions: $p^+ = P(e|y = 1)$ and $p^- = P(e|y = 0)$ are the true positive distribution and true negative distribution respectively; $\hat{p}^+ = P(e|s = 1)$ and $\hat{p}^- = P(e|s = 0)$ are the observable positive distribution and the observable negative distribution respectively; $\pi^+ = P(y = 1)$ and $\pi^- = P(y = 0)$ are the class priors of the ground truth; $p = P(e)$ is the edge prior, assumed to be uniform by default; $P(y = 1|e)$ is the value that we wish to obtain ultimately.

If the data is unbiasedly sampled then $\hat{p}^+ = p^+$, $\hat{p}^- = p^-$. We hold this assumption until Section 6 when we address the case of biased observation. Also notice that $G$ can either be viewed as containing a set of edges $E$ drawn from distribution $\hat{p}^+$, or equivalently as containing a set of negative edges, $V^2 \backslash E$, drawn from distribution $\hat{p}^-$.

**Difference between $q^-, p^-(\cdot|u), \hat{p}^-(\cdot|u)$.** It is crucial to distinguish the three negative distributions noted above: $q^-, p^-(\cdot|u), \hat{p}^-(\cdot|u)$, which are all defined over node set $V$. $q^-$ is the proposal negative distribution whose optimal form we seek to find, which may or may not depend on the anchor node $u$ though. That is why in the notation we suppress the dependence of $q^-$ on any other entities for the time being. $p^-(\cdot|u)$ is the true negative distribution given anchor node $u$ fixed, which we cannot directly observe if the observations are biased; $\hat{p}^-(\cdot|u)$ is the observed negative distribution, the distribution that our data (*i.e.* the observed negative edges) are directly sampled from. In fact, our ultimate goal in this paper is to construct $q^-$ from $\hat{p}^-(\cdot|u)$, as introduced in Section 1.

## 4 UNBIASED OBSERVATIONS

In this section, we show how $q^-$ can be derived based on the idea that negative sampling is essentially approximating an established risk term. The result in this section will also serve as the basis for discussion of the biased observation case in Section 6.

Given a graph $G = (V, E)$ with distributions $p^+$ and $p^-$ as defined in Section 3, consider the following theoretical risk term that we ultimately wish to minimize, associated with encoder $f$ with parameters $\theta$:

$$R_2(f) = \mathbb{E}_{(u,v)\sim p^+}[-\log p_f(v|u)] \tag{4}$$

$$\text{where } p_f(v|u) = \frac{e^{g(f(x_u), f(x_v))}}{e^{g(f(x_u), f(x_v))} + \tau \mathbb{E}_{v^- \sim p^-(\cdot|u)} e^{g(f(x_u), f(x_{v^-}))}} \tag{5}$$

where $p_f(v|u)$ is the generalized softmax probability of node $v$ given anchor node $u$, based on $f$'s output. See more discussion in Appendix B.2 for ensuring $p_f$ to be a proper distribution.

Crucially note that the negative distribution in the denominator of Eq.5 is $p^-$, instead of $q^-$. This is because we desire the probability of each edge to be as large as possible — calibrated against the probabilities of *all* other possible negative edges sampled from the *true* negative distribution $p^-$, rather than against edges from an arbitrary proposal distribution $q^-$. Using $p^-$ for calibration reflects the competition that positive edges face from the entire set (distribution) of negative edges, which represents the general objective we seek to optimize.

$R_2(f)$ can be further interpreted by writing down its empirical estimator:

$$\hat{R}_2(f) = \frac{1}{n} \sum_{\{(u_i, v_i)\}_{i=1}^n \sim \hat{p}^+} [-\log p_f(v|u)] = \frac{1}{|E|} \sum_{(u,v) \in E} [-\log p_f(v|u)] \tag{6}$$

which has been widely used in graph learning's literature. For example, when $\tau = 1$ this becomes the InfoNCE loss (Oord et al., 2018). Minimizing $\hat{R}_2(f)$ is equivalent to maximizing the probability of the following random graph model, assuming uniform node prior $P(u)$:

$$P(G) = \prod_{u,v \in V} P(u,v)^{p^+(u,v)} = \prod_{u,v \in V} [p_f(v|u)P(u)]^{p^+(u,v)} = Z \prod_{(u,v) \in E} [p_f(v|u)]^{\mathbb{1}_{\{(u,v) \in E\}}} \quad (7)$$

If our ultimate goal is to minimize $R_2(f)$, what we tell about the sampling problem defined in Section 3? The following proposition shows that there exists a unique choice of $q^-$ which ensures that optimizing the $R_1(f)$ in Eq. 4 is identical to optimizing the $R_2(f)$ in Eq. 3.

**Proposition 2.** $\nabla_\theta R_1(f) \equiv \nabla_\theta R_2(f)$ *if and only if both of the following conditions hold:*

- *For each $u \in V$ fixed, $q^-(v) \propto p_f(v|u)p^-(v|u)$*

- *$L^+ = -g, L^- = z_u g$, where $z_u = \mathbb{E}_{v^- \sim p^-(\cdot|u)}[p_f(v^-|u)]$*

*Proof.* See Appendix B.3. $\qquad\square$

Since $\nabla_\theta R_1(f) \equiv \nabla_\theta R_2(f) \Rightarrow \arg\min_\theta R_1(f) \equiv \arg\min_\theta R_2(f)$, this proposition crucially shows that for optimizing the risk term $R_2(f)$, the best proposal negative distribution $q^-$ to be used in Eq. 1 should take the simple form of $q^- \propto p_f(\cdot|u)p^-(\cdot|u)$. To exactly approximate $R_1(f)$, $g^-$ should be further reweighed by the partition $z_u$, which however does *not* need to be explicitly approximated as it cancels off with the $z_u$ for normalizing $q^-$. Also see Appendix B.3 for details.

To analyze the implications of the reweight term $p_f(v|u)$, first notice that the equivalence above is defined between the two gradients, meaning that the property holds throughout the training of $f$. In other words, $p_f(v|u)$ is a changing probability term in training – calculated based on the output of the model $f$ in the current training epoch. Also notice that a large $p_f(v|u)$ means that the current model $f$ leans towards classifying $(u,v)$ as positive. Therefore, using $p_f(v|u)$ to reweigh the observed (true) negative distribution $p^-$ is a precise form of *adaptive hard* negative sampling. This theoretical result also intriguingly relates in spirit to some of the previous heuristic-driven sampling methods for adaptive hard negative sampling: (Zhang et al., 2013; Robinson et al., 2021; Lai et al., 2024).

To implement the sampling method based on Proposition 2, we only need to approximate $p^-$ and then reweigh it by $p_f$ via importance sampling. To efficiently approximate $p^-$, we resort to the relationship $p = p^+\pi^+ + p^-\pi^- \Leftrightarrow p^- = (p - p^+\pi^+)/\pi^- = (p - \hat{p}^+\pi^+)/\pi^-$. This allows us to exploit the sparsity of observed $\hat{p}$ in practice. Refer to Section 3.2 for more definitions. Here, the positive prior $\pi^+$ is a hyperparameter to tune, and $\pi^- = 1 - \pi^+$. For approximating $\hat{p}^+$, it is typical to use the Monte Carlo method (Robert, 1999), which averages the delta function over all observed neighboring edges: $\hat{P}^+(\cdot|u) \simeq \frac{1}{|\mathcal{N}(u)|} \sum_{v \in \mathcal{N}(u)} \delta_{u,v}(\cdot)$, where $\delta_{u,v}$ is the Dirac delta function, and $\mathcal{N}(u)$ is the neighbors of $u$. Finally, $p_f$ can be approximated by replacing the expectation operator in its definition by samples average.

## 5   Biased Observations with Partially Observable Positives

In this section, we present a sampling method for biased graph observations with only partially observable positive edges. Section 5.1 motivates the bias setup. Section 5.2 introduces an empirical estimator and its corresponding sampling method. Section 5.3 further describes a graph-learning-based method for estimating a key term in the sampling method.

### 5.1   Bias Formulation

Many of the real-world graphs have this interesting property of observation bias: while the observation of an edge often signifies a reliably logged of interaction, the absence of an edge could either mean that the edge does not exist in reality, or that the existence of the edge just has not been testified.

For example, in recommendations, the absence of an user-item edge can either mean the user truly dislikes the item, or that the user just has not been offered the chance to interact the item but would have liked it if so. This also applies to other types of interactions such as social interactions and

protein interactions, and patient-disease diagnosis relations. This type of observation bias constitutes a significant source of noise in negative sampling.

To formulate this bias, recall from Section 3 definitions of the random variables: we consider probabilities in the sample space $\{(e, y, s)\}$ where $e \in V^2$ is a random node pair, $y \in \{0, 1\}$ is the true class variable, and $s \in \{0, 1\}$ is the observation indicator variable.

It is important to distinguish the meaning between $y$ and $s$. As an example in video recommendations, $e = (u, v)$ represents a user $u$ and a video $v$ that we consider. $y = 1$ means that $u$ *would have* liked $v$ if $u$ is offered the chance to view $v$. Also refer to the "ideal world" explanation in Section 1. $s = 1$ means the fact that $u$ likes $v$ is actually logged by the platform, usually by judging from some predefined interaction metrics by the platform such as the actual action of clicking on the "like" button, long video viewing time, positive comment below the video, *etc.*

The observation bias that we consider stipulates that $s = 1$ only when $y = 1$, or equivalently, $P(s = 0|y = 0) = 1$ by contrapositive. We further define $\phi_{u,v} = P(s = 1|y = 1, e = (u, v))$, which is the probability for edge $e$ to be observed given itself being a true positive. $\phi_{u,v}$ is also called propensity score in some literature.

## 5.2 Unbiased Risk Estimator Using Biased Observations

In this case, directly applying the sampling method over the observable distributions is problematic, because both the $p^+$ in the problem definition (Eq.3), and the $p^-$ in $q^- \propto p_f(\cdot|u)p^-(\cdot|u)$, are now unobservable. If we directly replace them by $\hat{p}^+$ and $\hat{p}^-$, further corrections are needed. In fact, we will show that in this case not only $\hat{p}^-$ needs to be corrected, but so does $\hat{p}^+$. In other words, in order to approximate $R_2(f)$ under biased observation, we need to build two proposal distributions on top of $\hat{p}^+$ and $\hat{p}^-$: a proposal *positive* distribution $q^+$, and a proposal negative distribution $q^-$. The following theorem elaborates this idea.

**Theorem 1.**

$$R_2(f) \equiv \mathbb{E}_{u \sim \hat{p}^+}[R_2(f, u)] \tag{8}$$

$$where \quad R_2(f, u) = \mathbb{E}_{v \sim \hat{p}^+(\cdot|u)}\left[\frac{c}{\phi_{u,v}}L^+\right] + \mathbb{E}_{v \sim \hat{p}^-(\cdot|u)}\left[\frac{(1 - \pi^+c)c}{\pi^-\phi_{u,v}}p_f(v)L^-\right]$$

$$+ \mathbb{E}_{v \sim \hat{p}^+(\cdot|u)}\left[\frac{\pi^+c(1-c)}{\pi^-\phi_{u,v}}p_f(v)(-L^-)\right] \tag{9}$$

*with $c = P(s = 1|y = 1)$, $\phi_{u,v} = P(s = 1|y = 1, e = (u, v))$, $\hat{p}^+, \hat{p}^-, \pi^+, \pi^-$ defined in Section 3.1, $p_f(v)$ defined in Eq. 17.*

*Proof.* See Appendix B.4. □

**Corollary 1.** *If $L^+ = -L^-$, then*

$$R_2(f) \equiv \mathbb{E}_{u \sim \hat{p}^+}\left[\mathbb{E}_{u \sim q^+}[\beta_1 L^+] + \mathbb{E}_{u \sim q^-}[\beta_2 L^-]\right] \tag{10}$$

*with the following assignment of proposal distributions $q^+$, $q^-$, and constant loss weights $\beta_1, \beta_2$:*

- *For each $u \in V$ fixed, $q^+(v) \propto \phi_{u,v}^{-1}(p_f(v|u) + \beta_1)\,\hat{p}^+(v|u)$*

- *For each $u \in V$ fixed, $q^-(v) \propto \phi_{u,v}^{-1}\,p_f(v|u)\,\hat{p}^-(v|u)$*

- *$\beta_1 = \frac{\pi^+c(1-c)}{\pi^-}$, $\beta_2 = \frac{(1-\pi^+c)c}{\pi^-}$*

*Proof.* See Appendix B.5. □

Corollary 1 directly gives the desired form for the new sampling method that best approximates $R_2(f)$. Similar to the unbiased case, here we also see $p_f$ show up as a reweight to the observed distributions, which carries rich implications – see previous discussion under Proposition 2.

To implement the sampling method, first notice that both $c$ and $\pi^+$ do not depend on $v$, and $\pi^+ c = P(s = 1)$ can be directly estimated from the data by $\frac{2|E|}{N(N-1)}$. $\pi^+$ remains to be a sampling hyperparameter to tune, and $\pi^- = 1 - \pi^+$. To efficiently approximate $\hat{p}^-$, we resort to the relationship $p = p^+ \pi^+ c + p^-(1 - \pi^+ c) \Leftrightarrow \hat{p}^- = (p - \hat{p}^+ \pi^+ c)/(1 - \pi^+ c)$, which allows us to exploit the sparsity of $\hat{p}$ in practice. Refer to Section 3.2 for more definitions.

The only unknown that depends on $v$ is the propensity score function $\phi_{u,v}$. We elaborate its estimation in the next subsection.

## 5.3  Learning-based Estimation of Propensity Scores Using Graph Features

In this subsection, we present a learning-based method for approximating the propensity function $\phi_{u,v}$. We will first discuss its interpretation, the we will introduce how it can be approximated by a graph encoder parameterized by a neural network.

**Interpretation.** $\phi_{u,v} = P(s = 1|y = 1, e = (u, v))$ is essentially the observation bias of edge $(u, v)$, also known as *exposure bias* or *labeling bias* in different literature. Here, our key observation is that the existing literature discussing these biases essentially establishes their strong correlation with various *graph-based features*, such as (1) Popularity bias (Zhang et al., 2021), which may be quantified by node degree $d_u, d_v$ (2) Position bias Chen et al. (2023), which may be quantified by the output of distance function $g$ on the node pair $(u, v)$. In addition, we also conjecture that the general structural information of the $u, v$ may also be correlated factors.

Although these bias terms can be easy to identify, conceptualize, and quantify, their relationship with the propensity score, as encapsulated by $\phi$, is very complex and varied from one domain to another. Since both the input and the desired output has been clear, here we propose to use a neural network as the universal function approximator for $\phi$.

**Parameterization.** We define the neural approximator for $\phi$ as

$$\hat{\phi}(u, v) = \sigma(\text{MLP}([d_u; d_v; f(u); f(v); g(u, v)]))  \tag{11}$$

where $\sigma$ is a non-linearity function such as sigmoid; the degree features and the embeddings of $u, v$ are concatenated as input to the MLP; $g(u, v) = g^+(f(u), f(v))$, see Sec.3.1. The output $\hat{\phi}(u, v)$ is directly plugged into Eq. 27 so that $\hat{\phi}$ is co-trained with $f$ in an end-to-end fashion. The role of $\hat{\phi}$ in Eq. 27 is very similar to that of the attention mechanism (Veličković et al., 2018; Vaswani, 2017): both are a plugged-in module that learns to reweigh samples (tokens) alongside primary training; the goal of both modules is also to approximate an underlying, physically motivated function.

**Regulation.** $\hat{\phi}$ can be further regulated to align its behavior with the desired properties of $\phi$. Besides the unit-range output enforced by $\sigma$, we also consider its first moment and consistency:

- First moment: $\mathbb{E}_{u,v \sim p^+}[\phi_{u,v}] = c$ ;
- Consistency: $g(u, v_1) > g(u, v2) \Leftrightarrow \phi(u, v_1) > \phi(u, v_2), \quad \forall u, v_1, v_2 \in V$.

The first-moment constraint essentially stipulates the rough numerical scale of $\phi$, *i.e.* to be centered around $c$ by expectation. See its proof in Appendix B.6. We softly enforce this by mean squared loss $(\hat{\phi}(u, v) - c)^2$. The consistency constraint is based on the probabilistic gap theory (He et al., 2018; Gerych et al., 2022), which states the general order-preserving property between the propensity function and the decision function. We encourage it by list-wise ranking loss ListMLE (Xia et al., 2008), computed between the two length-$(k + 1)$ lists, $(g(u, v^+), g(u, v_1^-), ..., g(u, v_K^-))$ and $(\hat{\phi}(u, v^+), \hat{\phi}(u, v_1^-), ..., \hat{\phi}(u, v_K^-))$.

## 6 EXPERIMENT

We validate the performance of our new sampling methods derived in Proposition 2 and Theorem 1 on real-world datasets. We also investigate the neural approximator for propensity scores developed in Section 5.2 via ablation studies.

### 6.1 EXPEARIMENTAL SETUP

**Baselines.** We compare with 8 representative negative sampling methods, including both classical and state-of-the-art methods: random sampling (RNS) (Rendle et al., 2012), popularity-based sampling (PNS) (Chen et al., 2017), random-walk-based sampling (RWNS) (Ying et al., 2018), DNS (Zhang et al., 2013)), MCNS (Yang et al., 2020b), SENSEI Yan et al. (2024), IRGAN Wang et al. (2017), NMNR Wang et al. (2018).

**Dataset.** We use three popular benchmarks for evaluating graph-based recommendations: MovieLens (Ding et al., 2020), Pinterest (Ding et al., 2020; Geng et al., 2015), and LastFM Wang et al. (2019). See Appendix C.1 for more details.

**Data Split.** Special care is needed when splitting the edge set into training, validation and testing. In principle, we should have $E_{\text{train}}, E_{\text{val}} \sim \hat{p}^+$ and $E_{\text{test}} \sim p^+$. However, in reality there is compromise and so we may only strive to approach this principle as best as we can.

We use the LastFM dataset for the unbiased scenario. Since $p^+ = \hat{p}^+$, the split can be done completely at random. Note that this underlyingly regards the entire given dataset as samples from the ground-truth $p^+$, even though this still may not hold in reality during data collection. The splits are $70\% - 15\% - 15\%$ for training, validation, and testing.

We use the MovieLens and the Pinterest dataset for the biased scenario. Since we have no direct access to $p^+$. we split the edges in temporal order by following the classical "leave-one-out" strategy in He et al. (2017); Rendle et al. (2012): leaving the most recent and the second most recent edge for each node as testing and validation set, respective.

**Base Model.** Note that the choice of the base model $f$ is independent from the sampling method. We use (1) a matrix-factorization based model (Rendle et al., 2012), which is essentially the base model of many classical node embedding methods such as deepwalk (Perozzi et al., 2014) and node2vec (Grover & Leskovec, 2016) and (2) a GNN-based model, LightGCN (He et al., 2020), which is a popular GNN model for recommendation.

**Configurations.** The configurations related to base models which are relatively simple and not the main focus of comparison. In this paper, we use Adam optimizer, learning rate $1e - 4$, L2 regularization $1e - 5$, hidden embedding size $64$, mini-batch size $1024$ with early stopping, negative sampling number $k = 16$, to tune the base models with different sampling methods. For our sampling method, we set $\tau = k = 8$; we further tune the class prior $\pi_+$ ranged from $5\%$ to $30\%$; we set $g^-$ as the dot product and $g^+ = -g^-$. For hyperparameters related to the baseline sampling methods, we follow their reported setting for tuning.

### 6.2 RESULTS AND ANALYSIS

**Performance under Biased Observations.** As shown in Table 1, our method consistently outperforms all baseline methods across both models and datasets in the biased observation scenario. Remarkabley on the Pinterest dataset with the LightGCN model, our method attains a Recall@20 of $17.61\%$, very significantly surpassing the next best method. The results demonstrate the effectiveness of our proposed sampling strategy in handling biased observations with partially observable positives.

**Performance under Unbiased Observations.** On the LastFM dataset, which simulates the unbiased observation scenario, our method achieves competitive performance compared to the baselines. We note that our method is extremely simple and that the competitive baseline DNS itself is also a highly flexible form of adaptive hard negative sampling.

| Sampling | MF (Rendle et al., 2012) | | | LightGCN (He et al., 2020) | | |
|---|---|---|---|---|---|---|
| | MovieLens | Pinterest | LastFM | MovieLens | Pinterest | LastFM |
| RNS | 7.11 ± 0.16 | 7.50 ± 0.19 | 5.97 ± 0.13 | 8.94 ± 0.11 | 7.98 ± 0.25 | 6.39 ± 0.10 |
| DNS | 12.42 ± 0.09 | 8.56 ± 0.06 | **7.37 ± 0.21** | 11.89 ± 0.08 | 10.77 ± 0.13 | **7.26 ± 0.12** |
| PNS | 8.41 ± 0.10 | 8.42 ± 0.05 | 6.64 ± 0.11 | 7.58 ± 0.05 | 8.67 ± 0.05 | 7.10 ± 0.09 |
| IRGAN | 10.57 ± 0.12 | 9.11 ± 0.09 | 6.86 ± 0.09 | 10.23 ± 0.25 | 9.79 ± 0.26 | 7.13 ± 0.11 |
| NMRN | 8.76 ± 0.19 | 7.67 ± 0.11 | 6.87 ± 0.11 | 10.22 ± 0.05 | 7.62 ± 0.06 | 7.12 ± 0.14 |
| RWNS | 9.27 ± 0.06 | 7.93 ± 0.05 | 6.59 ± 0.18 | 11.12 ± 0.08 | 8.00 ± 0.06 | 6.68 ± 0.22 |
| MCNS | 8.53 ± 0.08 | 8.68 ± 0.17 | 6.65 ± 0.20 | 7.22 ± 0.05 | 8.85 ± 0.11 | 6.88 ± 0.12 |
| SENSEI | 10.19 ± 0.06 | 9.01 ± 0.10 | 5.13 ± 0.25 | 8.10 ± 0.11 | 10.78 ± 0.14 | 5.44 ± 0.21 |
| **Our Method** | **12.77 ± 0.13** | **12.59 ± 0.12** | **7.38 ± 0.11** | **17.41 ± 0.19** | **17.61 ± 0.20** | **7.39 ± 0.12** |

Table 1: Performance comparison on real-world datasets (Recall@20, -%). We test 8 baselines over 2 base models over 3 different datasets. MovieLens and Pinterest have data splits to simulate biased observation; LastFM has data splits to simulate unbiased observation. In each case, only the corresponding version of our sampling method is tested.

## 6.3 ABLATION STUDIES

To understand the contribution of each component in our proposed method, we conduct ablation studies on the Pinterest dataset using the LightGCN model. We examine the impact of the following factors: (1) **Adaptive Reweighting ($p_f$):** We assess the effect of the adaptive reweighting term derived from the current model's output. (2) **Propensity Score Estimation ($\phi_{u,v}$):** We assess the importance of accurately estimating the propensity scores. (3) **Regularization on $\phi_{u,v}$:** We assess the effect of removing the regularization terms (first-moment and consistency constraints) on the propensity score estimator. (4) **Class Prior $\pi_+$:** We assess the sensitivity of the method to different values of the class prior $\pi_+$.

Table 2 summarizes the results of the ablation studies.

| Method Variant | Recall@20 |
|---|---|
| (A) Full method | 17.61 |
| (B) Without adaptive reweighting $p_f$ | 6.60 |
| (C) Constant propensity score ($\phi(u,v) = c$) | 12.39 |
| (D) Without regularization on $\phi(u,v)$ | 12.08 |
| (E) Varying class prior $\pi_+$ (0.01) | 13.33 |
| (F) Varying class prior $\pi_+$ (0.005) | 17.05 |
| (G) Varying class prior $\pi_+$ (0.001) | 4.92 |

Table 2: Ablation study on the components of our method on the Pinterest dataset using LightGCN (He et al., 2020) (Recall@20)

**Effect of Adaptive Reweighting.** Comparing the full method (A) with the variant without adaptive reweighting (B), we observe a significant drop in Recall@20 from 17.61% to 6.60%. This demonstrates the importance of the adaptive hard negative sampling strategy derived from Proposition 2.

**Impact of Propensity Score Estimation.** When we assume a constant propensity score (C), the performance decreases to 12.39% and 12.08%, respectively. This indicates that accurately estimating the propensity scores $\phi_{u,v}$ is crucial for correcting the observation bias in the data, as proved in Section 5.3. Similar to the attention mechanism, this estimation allows the model to account for the varying likelihood of positive edges being observed, leading to more reliable learning.

**Role of Regularization on $\phi_{u,v}$.** Removing the regularization terms on the propensity score estimation (D) results in a noticeable performance drop to 12.08%. This suggests that the first-moment constraint and consistency constraint introduced in Section 5.3 help improve the estimation of $\phi_{u,v}$ by enforcing desirable properties, thus enhancing the overall performance.

**Effect of Class Prior $\pi_+$.** We also examine the impact of varying the class prior $\pi_+$ (E–G). The best $\pi_+$ is 0.006. Setting $\pi_+$ to 0.005 yields a Recall@20 of 17.05%, which is close to the performance of the full method. However, setting $\pi_+$ too low (0.001) or too high (0.01) leads to suboptimal performance, with Recall@20 scores of 4.92% and 13.33%, respectively. This indicates that while our method is relatively robust to the choice of $\pi_+$ within a certain range, tuning this hyperparameter is important for achieving optimal performance. The class prior affects the weighting of positive and negative samples in the unbiased risk estimator, and an appropriate value helps balance the learning process.

Overall, the ablation studies confirm that each component of our method contributes to its effectiveness. The adaptive reweighting mechanism and the accurate estimation of propensity scores are particularly critical for achieving superior performance in graph-based recommendation tasks with implicit feedback.

## 7 CONCLUSION

In conclusion, in this work we shifted the paradigm of negative sampling in graph representation learning by focusing on approximating and leveraging the true negative distribution. We derived optimal sampling strategies for both unbiased and biased observed graphs, connecting them to maximum likelihood estimation and utilizing graph topology. Our methods showed significant improvements over existing approaches on real-world datasets, validating our theoretical contributions and offering a new direction for negative sampling strategies.

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

# Appendix

## A  IMPLICIT DATA ASSUMPTIONS OF EXISTING METHODS

Should we consider negative sampling strategies to be primarily a matter of human design, or does a ground-truth distribution of negative samples $p^-$ exist objectively and independently of our choices (and further that its exact form in the real world cannot be easily articulated by "principles" due to nature's complexity)?

For this question, an interesting observation is that many of the existing methods have implicitly taken taken an ambiguous or mixed stance.

To see this, notice that many previous works for *negative* sampling have left *positive* sampling untouched. This essentially respects the existence a ground-truth positive distribution in nature, from which the observed data (positive edges) get sampled unbiasedly. However, respecting the observed ground-truth positive distribution is equivalent to respecting its corresponding ground-truth negative distribution (by simply taking complement). This is because any graph in nature can be simultaneously viewed as a collection of positive edges sampled from the positive distribution, or a collection of negative edges sampled from the negative distribution. However, in the meantime those existing methods are proposing new negative distributions.

This seeming self-contradiction can only be explained when we consider the existing methods to have respected the existence of $p^-$, but in meantime be proposing a negative distribution different from their respected ground-truth for other good learning properties such as potentially faster empirical convergence. However, this advantage is at the compromise of unbiased estimation, and often does not have theoretical guarantee.

## B  PROOFS AND THEORETICAL DISCUSSION

### B.1  PROOF OF PROPOSITION 1

$$J = \sum_{(u,v)\in E,\, v_1^-,\ldots,v_k^-\sim q^-(v)} \left[ L^+(u,v) + \frac{\tau}{k}\sum_{i=1}^{k} L^-(u,v_i^-) \right] \tag{12}$$

$$\simeq \mathbb{E}_{(u,v)\sim p^+}\left[ L^+(u,v) + \tau \mathbb{E}_{v^-\sim q^-(v)} L^-(u,v^-) \right] \tag{13}$$

$$= \mathbb{E}_{u\sim p^+}\left[ \mathbb{E}_{v\sim p^+(\cdot|u)}\left[ L^+(u,v) + \mathbb{E}_{v^-\sim q^-(v)} L^-(u,v^-) \right] \right] \tag{14}$$

$$= \mathbb{E}_{u\sim p^+}\left[ \mathbb{E}_{v\sim p^+(\cdot|u)} L^+(u,v) + \mathbb{E}_{v\sim p^+(\cdot|u)}\left[ \mathbb{E}_{v^-\sim q^-(v)}\left[ L^-(u,v^-) \right] \right] \right] \tag{15}$$

$$= \mathbb{E}_{u\sim p^+}\left[ \mathbb{E}_{v\sim p^+(\cdot|u)} L^+(u,v) + \mathbb{E}_{v^-\sim q^-(v)} L^-(u,v^-) \right] \tag{16}$$

### B.2  GENERALIZED SOFTMAX

The full form of the generalized softmax is: for a selected edge $(u,v)$,

$$p_f(v|u) = \frac{e^{g(f(x_u),f(x_v))} + \tau p^-(v|u)e^{g(f(x_u),f(x_v))}}{e^{g(f(x_u),f(x_v))} + \tau \mathbb{E}_{v^-\sim p^-(\cdot|u)} e^{g(f(x_u),f(x_{v^-}))}} \tag{17}$$

$$p_f(v'|u) = \frac{\tau p^-(v'|u)e^{g(f(x_u),f(x_{v'}))}}{e^{g(f(x_u),f(x_{v'}))} + \tau \mathbb{E}_{v^-\sim p^-(\cdot|u)} e^{g(f(x_u),f(x_{v^-}))}}, \text{ for all } v' \neq v \tag{18}$$

This ensures that the probability distribution under the generalized softmax term in Eq. 5 is proper, because the denominator is not symmetric for all $v \in V$, *i.e.* the selected edge $(u,v)$ is treated differently than all other edge $(u,v')$ where $v' \neq v$. In practice since the support of $p^-(\cdot|u)$ usually does not encompass the positive node $v$ in the selected positive edge $(u,v)$, *i.e.* $p^-(v|u) = 0$, Eq. 17 degrades into Eq. 5.

## B.3 PROOF OF PROPOSITION 2

Denote $g(f(x_u), f(x_v))$ by $g_{uv}$

$$\nabla_\theta R_2(f) = \nabla_\theta \, \mathbb{E}_{(u,v)\sim p^+} \left[ -\log \frac{e^{g_{uv}}}{e^{g_{uv}} + \tau \mathbb{E}_{v^-\sim p^-(\cdot|u)} e^{g_{uv^-}}} \right] \tag{19}$$

$$= \mathbb{E}_{(u,v)\sim p^+} \left[ -\nabla_\theta g_{uv} + \nabla_\theta \log(e^{g_{uv}} + \tau \mathbb{E}_{v^-\sim p^-(\cdot|u)} e^{g_{uv^-}}) \right] \tag{20}$$

$$= \mathbb{E}_{(u,v)\sim p^+} \left[ -\nabla_\theta g_{uv} + \frac{e^{g_{uv}} \nabla_\theta g_{uv} + \tau \mathbb{E}_{v^-\sim p^-(\cdot|u)} e^{g_{uv^-}} \nabla_\theta g_{uv^-}}{e^{g_{uv}} + \tau \mathbb{E}_{v^-\sim p^-(\cdot|u)} e^{g_{uv^-}}} \right] \tag{21}$$

$$= \mathbb{E}_{(u,v)\sim p^+} \left[ -\nabla_\theta g_{uv} + \frac{e^{g_{uv}} \nabla_\theta g_{uv}}{e^{g_{uv}} + \tau \mathbb{E}_{v^-\sim p^-(\cdot|u)} e^{g_{uv^-}}} + \frac{\tau \mathbb{E}_{v^-\sim p^-(\cdot|u)} e^{g_{uv^-}} \nabla_\theta g_{uv^-}}{e^{g_{uv}} + \tau \mathbb{E}_{v^-\sim p^-(\cdot|u)} e^{g_{uv^-}}} \right] \tag{22}$$

$$= \mathbb{E}_{(u,v)\sim p^+} \left[ -\nabla_\theta g_{uv} + \mathbb{E}_{v^-\sim p^-(\cdot|u)} [p_f(v^-|u) \nabla_\theta g_{uv^-}] \right] \tag{23}$$

$$= \mathbb{E}_{u\sim p^+} \left[ \mathbb{E}_{v\sim p^+(\cdot|u)} [\nabla_\theta(-g_{uv})] + \mathbb{E}_{v^-\sim p^-(\cdot|u)} [p_f(v^-|u) \nabla_\theta g_{uv^-}] \right] \tag{24}$$

Notice that the first summation term is exactly positive sampling, and the second summation term is exactly negative sampling. Substitute $g^+ = -g_{uv} = -g(f(x_u), f(x_v))$, $g^- = z_u g(f(x_u), f(x_v))$ into the equation above:

$$\nabla_\theta \, R_2(f) = \mathbb{E}_{u\sim p^+} \left[ \mathbb{E}_{v\sim p^+(\cdot|u)} [\nabla_\theta L^+(u,v)] + \mathbb{E}_{v^-\sim p^-(\cdot|u)} \left[ \frac{1}{z_u} p_f(v^-|u) \nabla_\theta L^-(u,v^-) \right] \right] \tag{25}$$

where the partition $z_u = \mathbb{E}_{v\sim p^-(\cdot|u)}[p_f(v|u)]$. Meanwhile, from the definition of $R_1(f)$ in Eq. 4:

$$\nabla_\theta \, R_1(f) = \mathbb{E}_{u\sim p^+} \left[ \mathbb{E}_{v\sim p^+(\cdot|u)} [\nabla_\theta L^+(u,v)] + \mathbb{E}_{v^-\sim q^-} [\nabla_\theta L^-(u,v^-)] \right] \tag{26}$$

Therefore,

$\nabla_\theta R_1(f) \equiv \nabla_\theta R_2(f)$ if and only if $\forall u, v \in V, \quad q = \tau p_f(v|u) p^-(v|u)/z_u \propto p_f(v|u) p^-(v|u)$.

## B.4 PROOF OF THEOREM 1

Define $p = P(e)$ to be the edge prior. We first have the following propositions.

**Proposition 3.** $p^+ = \frac{c}{\phi_{u,v}} \hat{p}^+$.

*Proof.* $\hat{p}^+ = P(e|s=1) = P(e|y=1)P(s=1|y=1,e)/P(s=1|y=1) = p^+ \frac{\phi(e)}{c}$. $\qquad \square$

**Proposition 4.** $p^- = \frac{1}{\pi^-}(p - \pi^+ p^+)$.

*Proof.* $p = P(e) = P(e|y=1)P(y=1) + P(e|y=0)P(y=0) = \pi^+ p^+ + \pi^- p^-$. $\qquad \square$

**Proposition 5.** $p = \pi^+ c \hat{p}^+ + (1 - \pi^+ c) \hat{p}^-$.

*Proof.* $P(s=1) = P(s=1, y=1) = P(s=1|y=1)P(y=1) = \pi^+ c$, so $p = P(e) = P(e|s=1)P(s=1) + P(e|s=0)P(s=0) = \pi^+ c \hat{p}^+ + (1 - \pi^+ c) \hat{p}^-$. $\qquad \square$

Based on the above propositions, we have

$$R(f, u) = \mathbb{E}_{v \sim p^+}[L^+] + \mathbb{E}_{v \sim p^-}[p_f(v)L^-] \tag{27}$$

$$= \mathbb{E}_{v \sim p^+}[L^+] + \mathbb{E}_{v \sim (\frac{1}{\pi^-}(p - \pi^+ p^+))}[p_f(v)L^-] \tag{28}$$

$$= \mathbb{E}_{v \sim p^+}[L^+ - \frac{\pi^+}{\pi^-}p_f(v)L^-] + \mathbb{E}_{v \sim p}[\frac{1}{\pi^-}p_f(v)L^-] \tag{29}$$

$$= \mathbb{E}_{v \sim \frac{c}{\phi_{u,v}}\hat{p}^+}[(L^+ - \frac{\pi^+}{\pi^-}p_f(v)L^-)] + \mathbb{E}_{v \sim (\pi^+ c\hat{p}^+ + (1 - \pi^+ c)\hat{p}^-)}[\frac{1}{\pi^-}p_f(v)L^-] \tag{30}$$

$$= \mathbb{E}_{v \sim \hat{p}^+}[\frac{c}{\phi_{u,v}}L^+] - \mathbb{E}_{v \sim \hat{p}^+}[\frac{\pi^+ c}{\pi^- \phi_{u,v}}p_f(v)L^-] + \mathbb{E}_{v \sim \hat{p}^+}[\frac{\pi^+ c^2}{\pi^- \phi_{u,v}}p_f(v)L^-] + \mathbb{E}_{v \sim \hat{p}^-}[\frac{(1 - \pi^+ c)c}{\pi^- \phi_{u,v}}p_f(v)L^-] \tag{31}$$

$$= \mathbb{E}_{v \sim \hat{p}^+}[\frac{c}{\phi_{u,v}}L^+] + \mathbb{E}_{v \sim \hat{p}^+}[\frac{\pi^+ c(1 - c)}{\pi^- \phi_{u,v}}p_f(v)(-L^-)] + \mathbb{E}_{v \sim \hat{p}^-}[\frac{(1 - \pi^+ c)c}{\pi^- \phi_{u,v}}p_f(v)L^-] \tag{32}$$

## B.5 PROOF OF COROLLARY 1

Substitute definitions of $\beta_1, \beta_2, q^+, q^-$ into Eq. 1, we will see it exactly equals $R_2(f)$ in Theorem 1.

## B.6 PROOF OF THE FIRST-MOMENT CONSTRAINT IN SECTION 5.3

$\mathbb{E}_{u,v \sim p^+}[\phi_{u,v}] = \mathbb{E}_{u,v \sim p^+}[P(s = 1|y = 1, e = (u,v)] = \sum_{u,v \in V}[P(e = (u,v)|y = 1)P(s = 1|y = 1, e = (u,v)] = \sum_{u,v \in V}[P(s = 1, e = (u,v)|y = 1)] = \sum_e[P(s = 1, e = (u,v)|y = 1)] = P(s = 1|y = 1)x = c$

## B.7 PROOF OF THE CONSISTENCY CONSTRAINT IN SECTION 5.3

The probabilistic gap theory (He et al., 2018; Gerych et al., 2022) essentially states that $\phi(e) = h(P(y = 1|e) - P(y = 0|e))$, $h'(t) > 0$. Therefore, $\phi(e) = h(2g(e) - 1)$ and $\frac{d\phi}{dg}|_{t=e} = 2h'(g(e)) > 0$. Equivalently, $\phi(e_1) > \phi(e_1) \Leftrightarrow g(e_1) > g(e_1)$.

# C EXPERIMENT

## C.1 DATASET

The sources for the datasets used in this paper are:

- MovieLens (Harper & Konstan, 2015):
  https://grouplens.org/datasets/movielens/100k/.
- Pinterest (Geng et al., 2015):
  https://github.com/edervishaj/pinterest-recsys-dataset?tab=
  readme-ov-file
- LastFM: (Wang et al., 2019):
  https://github.com/xiangwang1223/knowledge_graph_attention_
  network

All datasets have been stated by their publishers to be freely available for research purpose.

