# OpenReview forum: "Truth-Guided Negative Sampling in Self-supervised Graph Representation Learning"
_ICLR.cc/2025/Conference — Submitted to ICLR 2025_

### Official Review · Reviewer_yKXA · 2024-10-26

**Soundness:** 3
**Presentation:** 3
**Contribution:** 4
**Rating:** 6
**Confidence:** 3

**Summary:**

This paper proposes a negative sampling approach from the ground up, offering a systematic analysis in contrast to other negative sampling studies. The paper also compares representative negative sampling methods and demonstrates significant improvements over previous approaches. However, there is room for improvement in the clearer definition of mathematical symbols (with some typos), the logical derivation of the sampling optimization objective, and the explanation of the sampling algorithm.

**Strengths:**

1. In contrast to other negative sampling studies, this paper reconsiders the formulation of negative sampling from the ground up, providing a systematic analysis. It begins with an ideal, unbiased estimator and then proposes a feasible optimization objective under biased conditions to design an effective sampling algorithm.
2. The paper compares representative negative sampling methods and shows a significant improvement in experimental results over previous approaches.

**Weaknesses:**

1. The paper contains numerous symbols, but some are either incorrect or poorly explained, increasing the reader's cognitive load. Details are elaborated in Question 1.
2. The implementation process of the sampling should be explained more clearly. While the current description aligns with the flow of the paper, it imposes a certain cognitive load on readers trying to understand the sampling algorithm. For instance, in Theorem 1, it is unclear how the class prior $\pi$  is defined, tuned, or calculated to advance the execution of the sampling algorithm.

**Questions:**

1. Typos and questions regarding symbols in the paper:
(1) In line 3 of page 2, is "$p^−= P(e∣y = 1)$" correct?
(2) In line 7 of section 3.2, "$\hat{p}^- = P(e∣s = 1)$", do you mean “$\hat{p}^+$” as the observable positive distribution?
(3) In equation 5, which is the function g, is it $g^+$ or $g^-$?
(4) What is the definition of $Z$ in equation 7? There seems to be an explanation for this symbol.
2.  What is the relationship between $R_1(f)$ and $R_2(f)$? What is the indication of Proposition 2?

---

> ### Author Response · Authors · 2024-11-28
>
> The authors would like to thank the reviewer for the review.
>
>
>
> **W1, Q1**
>
> Thanks for pointing out. We have fixed the typos in (1) and (2) in our revised version.
>
> For (3), function g is neither g^+ or g^-. g^+ and g^- are the empirical loss functions in negative sampling, while g is the function in the theoretical risk term (Eq. 5) that we ideally wish to minimize. Their relationship is shown in the condition 2 of Proposition 2, which basically says that negative sampling can be viewed as an approximation of the theoretical risk term, and one of the two premises for this to happen is that we need to assign g^+=-g, and g^-=z_u g. We will further clarify this in the text.
>
> For (4),  Z is simply a constant following the convention of math notations. We have clarified this in our revised version.
>
> **W2**
>
> The class priors $\pi^+$ and $\pi^-$ are defined in lines 175-176 in our revised version. Their calculation and tuning are explained in lines 325-326 and line 425.
>
> **Q2**
>
> The relationship and indication are explained in lines 224-226 and lines  234-238 in our revised version. Briefly speaking, this is the core idea of the entire Section 4:
> - R1(f) is the risk (loss) term that we empirically minimize when doing negative sampling
> - R2(f) is the risk (loss) term that we ultimately wish to minimize if we had infinite computing resource.
>
> Proposition 2 tells us that: if and only if we use its suggested design of the proposal negative distribution (q^-), will we be able to treat R1(f) as an unbiased approximation of R2(f). This result provides a theoretical ground for what should be considered a good proposal distribution.

---

### Official Review · Reviewer_thtV · 2024-10-28

**Soundness:** 2
**Presentation:** 2
**Contribution:** 2
**Rating:** 5
**Confidence:** 4

**Summary:**

This paper studies the problem that how to approximate and leverage the true distribution for better model performance. To address this issue, they derived optimal sampling strategies for both unbiased and biased observed graphs, connecting these strategies to maximum likelihood estimation and incorporating graph topology. The experimental results are conducted to demonstrate the efficiency of the proposed method.

**Strengths:**

1. The research problem is relevant.
2. The authors present a learning-based method for approximating the propensity function.
3. The authors conducted a comprehensive theoretical analysis.

**Weaknesses:**

W1. The authors need to thoroughly review their paper, as there are some errors that require revision. (See D1)

W2. The authors need to strengthen the experimental section of the paper. (See D2-D6)

**Questions:**

D1. In the Introduction section, the formula for deriving the negative sample distribution using Bayes' rule should be presented asp^-=P(e|y=0)∝(1-P(y=1|e))P(e) .

D2. The authors state that the observed distribution in the LastFM is the same as the true distribution, while this is not the case for MovieLens and LastFM. However, it is unclear how the authors determined this. Further clarification is needed to explain the reasoning or evidence behind this assertion.

D3. Lacking the necessary baselines from the past two years, such as [1], [2],[3]

[1] Huang T, Dong Y, Ding M, et al. Mixgcf: An improved training method for graph neural network-based recommender systems[C]//Proceedings of the 27th ACM SIGKDD Conference on Knowledge Discovery & Data Mining. 2021: 665-674.

[2] Shi W, Chen J, Feng F, et al. On the theories behind hard negative sampling for recommendation[C]//Proceedings of the ACM Web Conference 2023. 2023: 812-822.

[3] Lai R, Chen R, Han Q, et al. Adaptive hardness negative sampling for collaborative filtering[C]//Proceedings of the AAAI Conference on Artificial Intelligence. 2024, 38(8): 8645-8652.

D4. The authors should include additional ablation experiments to investigate the impact of different features, such as node degree features and node embeddings, on model performance when estimating the propensity function.

D5. The convergence speed is a crucial aspect of negative sampling methods. The authors should provide a comparison of the convergence rates between their method and the baselines.

D6. The authors should provide a time complexity analysis of their proposed method.

---

> ### Author Response · Authors · 2024-11-28
>
> The authors would like to thank the reviewer for the review.
>
> **D1**
>
> Thanks. We have fixed this typo in our revised version. We will also thoroughly review their paper to eliminate further typos.
>
> **D2**
>
> We would love to clarify that whether a dataset is biased or unbiased totally depends on the way we construct train/test splits. The datasets themselves only provide a set of recorded user-item edges. We explained how we conduct the splits differently for the biased and unbiased differently in Section 6.1.
>
> **D3**
>
> Thanks. We will follow the suggestions to add more experiments in our next version.

---

### Official Review · Reviewer_Yr1L · 2024-10-29

**Soundness:** 2
**Presentation:** 1
**Contribution:** 1
**Rating:** 3
**Confidence:** 5

**Summary:**

This work studies on unbiased sampling for graph representation learning and proposes a new method that leverages distribution transformation and propensity scores.

However, this work has some significant limitations in terms of novelty, presentation, I give a negative point of this work.

**Strengths:**

1. This work studies on an important problem.

2. Experiments on real-world datasets have been conducted to verify the efficacy of the proposed method.

**Weaknesses:**

1. My major concern lies on the novelty  Leveraging distribution transformation and propensity scores to overcome sampling bias is not a new concept and has been extensively studied in recent literature. Specifically, the relationship between $p^-$ and $p^+$ has been adopted by DCL [a5], and the same formula (line 1, page 4, in [a5]) is used to address sampling bias.  Furthermore, propensity scores have been widely used to address selection bias in the Recommendation Systems (RS) field [a6]. More critically, this work fails to review and explicitly cite these previous studies.

2. Another concern pertains to the clarity of the paper. Many important concepts are introduced without sufficient explanation, making the paper difficult to follow. Additionally, there are inconsistencies in definitions. For instance:

a) Eq.(4) is presented without much explanation. The motivation for introducing this formula and the reason for minimizing this objective remain unclear. The demonstration of unbiased estimation when using the distribution transformation of $p^-$ seems straightforward and does not require the introduction of these new concepts.

b) In section 3.2, $p^+$ is defined as $P(e|y=1)$, while in eq.(3), it is written as $p^+(.|u)$.  The interpretation of this notation is unclear. Meanwhile, eq.(7) introduces yet another notation, $p^+(u,v)$.

3. The experiments also have significant limitations:

a) Although this work claims to study samplers for graph representation learning, the experiments focus solely on recommendations. Notably, the selected baseline, MF, is not a graph-based method. Therefore, I suggest the authors conduct additional experiments on other graph learning scenarios and datasets.

b) The datasets used are relatively small. Larger datasets, such as Amazon, should be employed.

c) The baselines are outdated. Except for SENSEI, all the baselines predate 2020. More state-of-the-art sampling strategies could be employed, e.g., [a1][a2][a3][a4].

[a1]  IJCAI’24: Supervised contrastive learning with hard negative samples

[a2] WWW’23: Fairly adaptive negative sampling for recommendations

[a3] WWW’23: On the theories behind hard negative sampling for recommendation

[a4] CIKM’23: Batch-Mix Negative Sampling for Learning Recommendation Retrievers

[a5] NIPS’20: debiased contrastive learning

[a6] ICML’16: Recommendations as treatments: Debiasing learning and evaluation

**Questions:**

Please refer to weaknesses.

---

> ### Author Response · Authors · 2024-11-28
>
> The authors would like to thank the reviewer for the review.
>
> **W1**
>
> > the relationship between $p^-$ and $p^+$ has been adopted by DCL [a5], and the same formula (line 1, page 4, in [a5]) is used to address sampling bias.
>
> Our contribution towards addressing is way beyond this simple flipping trick: we have never claimed a contribution in connecting p^- and p^+ via the law of total probability, e.g. line 1, page 4, in [a5]. In fact, this flipping trick is a very well-established, simple result that has been seen in many of our cited works, including (Gerych et al.2022) and (Robinson et al. 2021) which is an improved version of [a5] written by the same authors.
>
> More importantly, the “bias” in [a5] is a totally different concept than the “bias” formulated in our paper (Section 5.1). The former simply refers to the issue that we have not excluded known positive samples when doing negative sampling. The latter refers to a much more challenging issue that we only have biased, partial observations of positive distribution which makes the estimation of negative distributions biased as well.
>
> > propensity scores have been widely used to address selection bias in the Recommendation Systems (RS) field [a6].
>
> Our method falls into the general category of Inverse Propensity Scoring (IPS) but has the following contributions and novelty:
>
> - Previous propensity methods are mainly designed for reweighing **observed** data points in recommendations. However, they have not been combined with negative sampling as a way to derive the optimal negative distribution in an end-to-end manner, which is what this paper is doing.
> - We use a very different method to estimate the propensity scores. Existing works such as [a6] estimates propensities via naïve bayes or regression over node attributes. Instead, we estimate propensity scores by novely exploiting its known connections various biases encapsulated by graph structural features, including node degrees and embeddings. This has been explained in the second paragraph of Section 5.3.
>
> We will add more references and discussion related to the propensity score methods.
>
>
> **W2**
>
> 2a) We explained the interpretations of Eq. (4) in lines 213-222, which show that Eq. (4) has intuitive connections with the famous InfoNCE loss and random graph model. Therefore, it is a well-motivated risk term that we ultimately desire to minimize.
>
> 2b) We defined and explained them in lines 162-164: an edge e contains two end nodes u, v, so p^+(e)= p^+(u, v)=P(e|y=1)= P(u, v|y=1), according to the definition of p^+. Because the second expression p^+(u, v) in this chain is essentially a joint distribution in space V \times V, we can define its corresponding conditional distribution p^+(\cdot|u).
>
>
> **W3**
>
> Thanks for your suggestions. We will follow them to improve our experimental section in the next version.

---

### Official Review · Reviewer_jryt · 2024-11-03

**Soundness:** 3
**Presentation:** 2
**Contribution:** 3
**Rating:** 5
**Confidence:** 3

**Summary:**

This paper studies the problem of negative sampling. It proposes to approximating and utilizing the true distributions in sampling methods for graph representation learning. The authors theoretically derive and analyze the forms that the optimal sampling strategy should follow, under the assumption that the positive edges are sampled unbiasedly from the true distribution, or that the positive edges are only partially observable. Experiments are conducted on recommendations.

**Strengths:**

1.This paper introduce a novel approach to negative sampling by proposing to approximate the true negative distribution This method provides a more principled approach to understanding and executing negative sampling.
2: This paper outlines a strategy to adjust the empirical risk estimator to account for bias in recommendation systems, enhancing the model’s applicability and robustness.

**Weaknesses:**

1: The format does not follow the formatting instructions. There is no line numbers, the captions are putted below table, the title in pdf does not align with that in the system. A lot of typos exist, e.g., in page 8, “respective”, and in table 1, why the results of DNS are bolded? The presentation makes the paper a little bit hard to follow.

2: It is not clear why experiments are connected on recommendation. The most recent baseline in this paper is SENSI [1], it conducts experiments on normal node classification tasks, using datasets like Cora, CiteSeer and PubMed.

3: The experiments lack an analysis of time and space complexity. Excessive time consumption may render this method impractical for use in real-world recommendation systems. No time complexity or runtime comparison. No pseudo-code and no data statistics. There are too few experimental metrics. The dataset for the ablation study is insufficient.

[1] Reconciling competing sampling strategies of network embedding

**Questions:**

1: Can the author investigate in detail why the proposed method can improve the performance, for example from the perspective of false negative rate or hard negative samples? Currently we only have one metric, Recall@20, and no quantitative analysis.  Even if there is no quantitative analysis, it is better to have some other metrics. As far as I know, it is not common to use only recall@k and fix k.

2. Can negative samples represent the entire non-interaction space?

---

> ### Author Response · Authors · 2024-11-28
>
> The authors would like to thank the reviewer for the review.
>
> **W1**
>
> Thanks. We have added line numbers and fixed the typos in “respective” and caption positions.
>
> For the results of DNS and title alignment:
> - The results of DNS are bolded because statistically they are also considered the best performance i.e.  close to our performance within the standard error.
> - The titles in the pdf and in the system do align. Could you please let us know what difference you have spot on them in reviewer’s portal?
>
> **W2**
>
> SENSEI did not conduct experiments on node classification tasks, but instead tested only on link-level tasks, including recommendation. Its experiment section states that “we evaluate the effectiveness of the proposed algorithm (SENSEI) for solving *link prediction* and *node recommendation* simultaneously in plain networks.”
>
> In fact, all the baselines we compared with have primary study link-level tasks, especially recommendations (link scoring and ranking).
>
> **Q1, W3**
>
> Thanks for the suggestion. We will supplement more experiments on this in our next version.
>
> **Q2**
>
> We explained this in lines 60-67. It is certainly most desirable if our negative samples can represent the entire non-interaction space, i.e. we are able to obtain sufficient amount of unbiased samples from ground-truth negative distribution p^-. However, in practice, because the entire non-interaction space is too large (N^2 where N is the number of nodes), the learning variance is very large too, which leads to underperformance. This is exactly why we need to modify the distribution of the negative samples, e.g. via boosting or hard negative sampling. From this perspective, our Section 4, especially Proposition 2, essentially shows how to modify the negative distribution in a theoretically-sound manner.

---

### Meta-Review · Area_Chair_3b8Q · 2024-12-20

**Metareview:**

The authors propose a negative sampling method for self-supervised graph learning. Their key idea is to leverage the underlying data distribution, and specifically, they derive optimal sampling strategies for both unbiased and biased training edge cases. The effectiveness of these strategies is demonstrated through empirical evaluations.

Most reviewers acknowledged:
- The importance of the considered problem
- The empirical effectiveness of the proposed method
- The comprehensiveness of the provided analysis

However, the reviewers also shared the following critical concerns:
- The presentation and methodological rigor need improvement
- More extensive experiments are required, covering a broader range of tasks, datasets, and baselines

Additionally, some reviewers highlighted the lack of complexity analysis and scalability tests.

Despite its clear merits, the paper has significant room for improvement.
Therefore, the meta-reviewer recommends rejecting the paper.

**Additional Comments On Reviewer Discussion:**

Major concerns remain unaddressed even after the discussion period.

---

### Decision · Program_Chairs · 2025-01-22

Reject